# Adverse Childhood Experiences on Reproductive Plans and Adolescent Pregnancy in the Gulf Resilience on Women’s Health Cohort

**DOI:** 10.3390/ijerph18010165

**Published:** 2020-12-28

**Authors:** Megan Flaviano, Emily W. Harville

**Affiliations:** 1School of Medicine, Tulane University, 1430 Tulane Avenue, New Orleans, LA 70112, USA; 2Department of Epidemiology, Tulane University School of Public Health and Tropical Medicine, 1440 Canal St. #8318, STE 2000, New Orleans, LA 70112, USA; eharvill@tulane.edu

**Keywords:** adverse childhood experiences, reproductive plan, adolescent pregnancy

## Abstract

We investigated if adverse childhood experiences (ACEs) and ACE sub-types were associated with increased odds of planning to have children and adolescent pregnancy. The Gulf Resilience on Women’s Health (GROWH) is a diverse cohort of reproductive-age women living in southeastern Louisiana during the 2010 Deepwater Horizon oil spill. In our sample of 1482 women, we used multinomial logistic regression to model odds ratios of wanting future children and assessed effect measure modification by educational attainment. We also estimated odds ratios of adolescent pregnancy with binomial logistic regression. Exposure to ACEs increased odds of wanting future children across all ACE sub-types. Among women with lower educational attainment, three or more ACEs (overall, childhood, and adolescence) had over two times the odds of wanting future children. History of ACE and the various sub-types, except for emotional abuse, were associated with increased risk of adolescent pregnancy. ACEs may be linked to adolescent pregnancy and reproductive plans, and variations by educational status highlighted social discrepancies and importance of social context in evaluation and intervention.

## 1. Introduction

Adverse childhood experiences (ACEs) are linked to poor medical outcomes and health behaviors [1,2]. For female reproductive health, ACEs have been associated with adverse birth outcomes [3,4,5], unintended pregnancy [6,7,8,9], early pubertal timing [10,11], and risky reproductive health behaviors, including early sexual debut and prenatal substance use [12,13,14]. Different kinds of ACEs, like physical abuse or sexual abuse, have shown varying associations with maternal health and psychosocial outcomes during pregnancy [15].

Adolescent pregnancy has a known association with ACEs [16,17,18], and childhood maltreatment, particularly sexual abuse, has been shown to raise risk of adolescent pregnancy [19]. Early life adversities have an additive association with age at first pregnancy [20]. A previous study on fertility-timing in impoverished areas of U.S. cities showed that high rates of teen childbearing bearing, severe uncertainty about future health, and excess mortality were more likely to occur in low-income and urban environments compared to nationwide averages, and it was theorized that adolescents, by conscious decision making or underlying social norms, were influenced to have children at a younger age to mitigate early health deterioration [21]. Teenaged pregnancy itself comes with maternal and child health risks of great public health concern, such as delayed prenatal care, low birthweight, preterm birth, stillbirth, preeclampsia, maternal depression, and neglected or delayed educational goals [22]. Unintended pregnancies during adolescence may be reduced with early interventions [22,23], yet the effect of ACE sub-categories on adolescent pregnancy could be explored further to better identify girls in need of this support.

ACE exposures could further impact adult reproductive intentions. Early psychological distress is theorized to influence the timing of reproductive plans [24]. Similar to the influence on adolescent pregnancy, accelerated life history- i.e., lower age at first birth- has been demonstrated in families living in neighborhoods short on economic resources, compared to families in more affluent areas, as a response to shorter life expectancies from low-income, unhealthy environments [25,26]. In comparison, experiences of neglect, like having to raise younger siblings, lowered pregnancy desire among Black women in a qualitative family planning study [27]. A study on HIV-infected women found childbearing motivations- shaped by negative early life experiences- were not linked to fertility intentions [28]. We have yet to ask if ACEs, and which kind of ACEs, are associated with women’s conscious intentions to have children before becoming pregnant.

How past traumas affect reproductive health can yield important information to better care for female patients across life stages in a trauma-informed care approach [29,30,31]. Because family planning is a routine part of healthcare provider-patient conversation [29,30,32], identifying psychosocial influences behind reproductive health plans may facilitate comprehensive assessments, leading to more impactful treatment or preventive measures [29,30]. Therefore, we investigated if ACEs and sub-types of ACEs, by age of exposure and specific categories of abuse, are associated with future reproductive plans and adolescent pregnancy. We hypothesized that ACEs would increase odds of wanting future children and adolescent pregnancy.

## 2. Materials and Methods

The Gulf Resilience on Women’s Health (GROWH) study is a cohort comprised of women living in southeastern Louisiana during the Deepwater Horizon oil spill in 2010. As part of a response to the oil spill, Project 1 of GROWH studied environmental exposures and their effects on reproductive-age women. Recruitment spanned 2011 to 2016, totaling 1788 women ages 18–45. Participants were recruited from health facilities offering OB/GYN services; Women, Infants, and Children (WIC) clinics; day care centers; and local community events. At recruitment, women were interviewed and completed an in-person questionnaire. Most questionnaires were completed at the site, but a few women completed them at home to mail back to investigators. 1620 women filled out at least 1 questionnaire or interview, including 443 women who were pregnant at the time of the interview. Blood and saliva samples were additionally requested [33,34].

For this study, one sample was used to analyze two outcome measures: reproductive plans and adolescent pregnancy (Table A1). From 1717 participants who had questionnaire data, the study sample (n = 1482) only included participants who were not missing values for reproductive plans, adolescent pregnancy, ACEs and sub-categories of ACEs (childhood, adolescence, physical, emotional, substance abuse, sexual, neglect), education, and race. There were no differences in frequency of ACE exposures and outcomes between included and excluded women in both reproductive plans and age at first birth samples. More excluded participants, relative to included participants, had high educational attainment and current depressive symptoms, and the excluded group was composed of more white, nulligravid and primigravid women. (Table A1). This study was approved by the Institutional Review Board on 3 October 2011 (reference number 239911). All participants provided written informed consent.

Adverse childhood experiences were evaluated via interview. Adverse childhood experiences (ACE) were measured through the Adverse Childhood Experiences Survey, which was adapted from the Family Health History Questionnaire create by Kaiser Permanente, in conjunction with the Centers of Disease Control and Prevention (CDC), for the Adverse Childhood Experiences Study [35]. The GROWH survey version assessed various hardships experienced during childhood (before age 12) and adolescence (between ages 12 to 17), including physical abuse, sexual abuse, emotional abuse, substance abuse, and neglect. ACE exposure was defined in general (“Overall ACE”) as well as ACEs classified by age (childhood, adolescence) and ACE sub-types (physical abuse, emotional abuse, substance abuse, sexual abuse, neglect). For overall, childhood, adolescence, physical, emotional, and substance ACEs, participants were divided into 3 exposure levels: “None” (0 ACEs), “Low” (1–2 ACEs) and “High” (≥3 ACEs). Sexual and neglect ACEs were separated into only 2 exposure levels due to sample size: “None” (0 ACEs) and “+1 ACEs” (≥1 ACEs).

For the reproductive plans outcome, a question was adapted from the National Survey of Family Growth [36]. Interviewees were asked, “Now, I would like to know your feelings about having a/another baby, whether or not you are able to, or plan to have one. Looking to the future, do you, yourself, want to have a/another baby at some time? [or] If it were possible, would you, yourself, want to have a/another baby at some time in the future?” Participants responded either “Yes”, “No”, or “Not Sure”. For the adolescent pregnancy outcome, women reported the age during their first pregnancy. Women were younger than 18 years old during their first pregnancy were categorized as “Yes”. Women who had their first pregnancy at 18 years old or older, and women who had never been pregnant fell under “No”.

Our covariates consisted of demographics: age (18–25, >25–30, >30–35, >35); race (black, white, neither black or white); highest educational attainment (high school or below, above high school); total annual household income (<$15,000, $15,000–<$35,000, ≥$35,000); and marital/cohabitation status (married, living with partner, separated/widowed/never married). Measures of obstetric history were added: current pregnancy status (pregnant or non-pregnant at time of interview); gravidity (nulligravid, primigravid, two pregnancies, three or more pregnancies); parity (nulliparous, parous); history of miscarriage (yes, no), and previous trouble getting pregnant in the past (yes, no). Self-reported mental health measures were also included- current depressive symptoms (yes, no as scored on the Edinburgh Depression Scale) [37] and PTSD-like symptoms (yes, no, as scored on the Post-Traumatic Checklist) [38] because of their significant associations with both overall ACE exposure and the adolescent pregnancy outcome.

For analyses, characteristics of all women in our study were first described, including prevalences of ACE categories and reproductive plans. Associations between ACE exposures and potential covariates were evaluated. Bivariate analyses were run to estimate the association between ACE exposures and outcomes. For estimating adjusted odds ratios, covariates associated with both exposure and outcome (*p* > 0.20) and covariates that theorized to bias associations between ACE exposure and outcomes were included [39,40]. Therefore, age, race, educational attainment, marital/cohabitation status, income, current pregnancy status, gravidity, parity, and previous trouble getting pregnant in all models were controlled for. The models for the reproductive plans outcome were also adjusted for history of miscarriage and previous trouble getting pregnant. For the adolescent pregnancy outcome, depressive and PTSD-like symptoms were adjusted in these models. As it might be a concern that women’s recent exposure to the oil spill would affect their health or plans, this was examined as a covariate. Adjusting for oil spill exposure, there were no differences in associations between crude and adjusted models, and only 12% of women had the highest oil exposure; therefore, oil spill exposure was not included as a covariate. Lastly, all covariates in our models were tested for statistical interaction with overall ACE using Wald tests. If effect measure modification was present (*p* < 0.05), then associations stratified by the effect measure modifier were estimated.

All unadjusted and adjusted odds ratios and 95% confidence intervals were estimated using binomial logistic regression for the adolescent pregnancy outcome and multinomial logistic regression for the reproductive plans outcome to model the three outcome types of no inherent order (“Yes ”, “No ”, “Not Sure”). Statistical significance was assessed at *p* < 0.05 for odds ratios and effect measure modification.

395 (27%) of women were missing data on covariates. Multiple imputation was used to address missing covariate data, using PROC MI and MIANALYZE in SAS. Multivariate normal distribution imputation algorithm was used to create 10 imputations. Variables used in the imputation model were all model covariates, continuous variable age, and history of depression counseling. The imputed sample was used for adjusted analyses. Statistical analyses were conducted using SAS University Edition 3.8 (SAS Institute Inc., Cary, NC, USA).

## 3. Results

Table 1 shows the demographic characteristics of the GROWH cohort sample population (n = 1482). Women were about evenly divided between those with negative (“No”, n = 735, 49.6%) reproductive plans, and those with positive (“Yes”, n = 640, 43.2%) or uncertain (“Not Sure”, n = 107, 7.2%) reproductive plans. “Yes” to wanting future children group had the largest proportion of 18 to 25-year-old women. The “Not Sure” group had the largest percentages of currently pregnant women and those with history of miscarriage. “No” to wanting future children group had the largest proportion of women older than 35 years old, history of 3 or more pregnancies, and parous women. In the total sample, 18.6% experienced a pregnancy before 18 years old. Women with history of adolescent pregnancy were more likely to have lower educational attainment and a history of 3 or more pregnancies.

Among each ACE sub-type, the most common ACE-subtype exposure level was None (Table 2). However, when ACEs were compiled together, a majority (52.0%) of women in the reproductive plans sample were exposed to three or more ACEs (Table 2). When separated by life stage period, most participants experienced at least one childhood and adolescent ACE (Table 2). Results from unadjusted analyses were organized in Appendix A.

Exposure to overall ACE and all ACE sub-types showed significantly higher adjusted odds of reporting “Yes” to wanting future children, which further increased from low to high ACE exposure, compared to women with no exposure in their respective ACE categories (Table 2). Neglect and sexual abuse were associated with lower adjusted odds of being “Not Sure” about wanting future children, compared to women with no neglect or no sexual abuse history (Table 2). Higher adjusted odds of being “Not Sure” increased with greater exposure levels of childhood, adolescent, and physical abuse ACEs, compared to those with no exposure to these ACEs (Table 2). Low emotional abuse and low substance abuse exposure, but not higher exposure levels, showed increased adjusted odds of reporting “Not Sure” (Table 2).

For the adolescent pregnancy outcome, increasing exposure to overall ACE, childhood ACEs, and adolescent ACEs were associated with increasing adjusted odds of adolescent pregnancy, compared to women without any ACEs (Table 2). History of sexual abuse, neglect, and high levels of physical abuse and substance abuse ACEs showed increased adjusted odds of adolescent pregnancy, compared to those with no exposure in the respectively ACE sub-category (Table 2). There was no effect measure modification by race or any covariate.

Overall ACEs differed between educational attainment groups in the sample. There was a statistically significant interaction between overall ACE exposure and highest educational attainment, as well as high childhood ACE and physical abuse exposure (*p* < 0.01) (Table 3). Among women with lower educational attainment, those with history of high overall ACEs, childhood ACEs, and adolescent ACEs had over two times higher odds of wanting future children, relative to respondents with low educational attainment not exposed to these ACEs (Table 3). Any level and type of ACE exposure showed increased adjusted odds of reporting “Yes” to reproductive plans among women with lower educational attainment (Table 3). In comparison, among women with higher educational attainment, those exposed to high overall and high childhood ACE levels, but not low, had higher odds of wanting future children, compared to women without respective ACE exposure (Table 3). Whereas low physical abuse exposure had increased adjusted odds of wanting future children among the lower educational attainment group, low physical abuse exposure for the higher educational attainment group was no longer associated with wanting future children (Table 3). Education-stratified odds ratios of “Not Sure” to reproductive plans showed no significant associations (Table A2).

## 4. Discussion

We hypothesized that adverse childhood experiences (ACEs) and sub-classifications of ACEs would be associated with increased odds of future reproductive plans to have children and adolescent pregnancy history. Exposure to ACEs were positively associated with wanting future children with effect measure modification by educational attainment. Those with sexual abuse and neglect history were less likely to be uncertain about wanting future children. Adolescent pregnancy showed a strong positive association with overall ACE and most ACE sub-types, except emotional abuse. Our results suggest that a history of ACEs may have implications for possible adolescent pregnancy and family planning intentions among certain patient populations.

As a component of reproductive health, intentions to have children could be affected by ACEs. Studies found that negative childhood experiences either increased drive to reproduce young in economically deprived environments [20], lowered pregnancy desire [27] or had no effect on pregnancy intentions among HIV positive women [28]. We found that self-reported exposure ACEs were strongly associated with increased intentions to have future children in adjusted analyses. On a systemic level, living in economically under-resourced neighborhoods can decrease healthy life expectancy, and which can increase intentions to have children at a younger age while relatively healthy [26]. A negative childhood environment can have damaging psychological repercussions, such as hopelessness, linked to greater pregnancy desire among female youth [41]. Childhood violence similarly can lead to increased risk of intimate partner violence among young women, who are vulnerable to reproductive coercion by their partners [42]. On the other hand, women may have a greater desire to start a family and to give their children a better and safer life, in contrast their own childhood experiences [42].

Not every type of ACE was related to wanting future children in our study. History of sexual abuse and neglect were associated with lower odds of being unsure about reproductive plans. Smigiel found that childhood sexual abuse was not linked to pregnancy desire among adolescents [43]. Holliday et al. did find that neglect impacted pregnancy intention; their study used a different approach involving thematic analysis of semi-structured interviews with 44 participants [27]. The type of ACE may matter when studying relationships between early hardships and reproductive health measures, along with their clinically relevant impacts on future health.

Future reproductive plans were influenced by different combinations of educational attainment level and type of ACE. Educational attainment was previously shown to interact with childhood sexual abuse to predict greater likelihood of having children [44]. We found an interaction between educational attainment and other types of ACEs (overall, high childhood ACEs, and low physical abuse) to predict future reproductive plans. Education level has been positively linked to reproductive plans, specifically in terms of fulfilling childbearing intentions [45,46]. However, we noted the lower educational attainment group, compared with the higher educational attainment one, had a larger effect estimate size of increased odds for wanting future children with history of ACEs. In previous studies, ACEs were associated with lower educational attainment [47,48]. ACEs also associated with increased unemployment [49] and lower occupational prestige or less-skilled jobs [47], and educational attainment attenuated both these associations [47,49]. Early life disadvantages were associated with poor adult mental health, which strengthen among those with low cognitive ability [50]. Not pursuing post-secondary education has been associated with increased desire for pregnancy and lowered disinclination to become pregnant among young, unmarried women [51].

One possible process tying these associations together could be through the perspective of Bachrach and Morgan’s social-cognitive model [52]. Fertility intentions are influenced by schematics of childbearing and parenthood formed from previous experiences, by position within societal structures, like educational status, and by demands of a person’s current situation [52]. The current situation for women, who have lower educational attainment, poor mental health impeding work ability, and childbearing ideals, all of which influenced by exposure to adverse childhood experiences, may have limited options to pursue careers, and therefore will more likely have and raise children rather than work. Bridger and Daly found that their associations between early life disadvantages and adult mental health were no longer observed with higher cognitive ability [50]. Women with ACE history but also higher cognitive ability to achieve higher educational attainment could, at least, develop fewer mental health issues hindering job prospects, and thus have more current job options to pursue careers. Effect measure modification by education showed its possible key influence on associations between ACEs and reproductive planning and further emphasized the importance of patient social histories to help identify those who may need support to fulfill intentions to have children.

In agreement with current literature, ACEs, sexual abuse, and most other types of ACEs were associated with increased adolescent pregnancy in the GROWH sample. ACEs are linked to teenage pregnancy [16,17,18]. Only emotional abuse ACEs were not associated with adolescent pregnancy in our study but have been shown to be linked to involvement in teenage pregnancy in previous study surveying of Minnesota high school students [18]. Childhood sexual abuse is associated with teenage pregnancy [19,53]. Rape carries the risk of getting pregnant [54] and upon discovery of sexual abuse, current pediatric practice guidelines include management of potential pregnancy [55]. Childhood sexual abuse may also affect cognitive development and lead to social or emotional cognitive distortions, which can impact sexual decision making and sexual activities and thus increase susceptibility to teenage motherhood [56]. Early sexual debut has been associated with ACEs, more so among women than men, and more often among women who have sex with women compared to heterosexual women [12]. Regarding sexual health education, in a systematic review by Zapata et al., contraceptive counseling lowered odds of unintended pregnancy among adolescents [23], which would be interesting to explore in future analyses. A National Longitudinal Study of Adolescent to Adult Health study described how decreased likelihoods of teenage pregnancy were linked to feeling close to others at school, earning a high school diploma, higher education enrollment, community service participation, or living in a two-parent home [57]. The lack of these resources in economically deprived environments, which have lower healthy life expectancies, may have strong influence on a younger at age first birth and teenage childbearing [21,25].

The GROWH cohort captures an array of general and reproductive health outcomes on a large, diverse group of women. GROWH women are representative of a target age group for child-bearing and reproductive-related questions. Diversity is a major strength in the cohort with about equal representation among Black and non-Black women. Self-reported reproductive plans are linked to future reproduction, albeit imperfectly [58,59]. To create the adolescent pregnancy variable, we used self-reported age at first pregnancy, because self-reported maternal age has been established as reliable as medical records [60].

Because we performed a cross-sectional analysis, we were limited to measuring future pregnancy intentions, instead of following a cohort to see if intentions aligned with actual reproductive outcome in the future. We lacked data on more proximate issues, such as contraceptive use, sexual health education history, age of sexual debut, sexual orientation, childhood and current family size, and current sexual activity. Negative reproductive plans in non-pregnant women combined with use of family planning methods are predictive of lower likelihood of pregnancy [57]. Desired family size is positively linked to the size of the family an individual has grown up in [61], which could influence reproductive decision making. Confounding by unmeasured factors such as current stress levels, employment status, and partner fertility intentions is possible. Participants included in the study sample did differ from excluded participants by education, race, gravidity, and depressive symptoms, which may introduce some bias into our results. Differentially underreported ACEs are a possible concern between different demographics, because lower numbers of ACEs were reported by Black participants compared to non-Black participants, and also by participants with lower educational attainment relative to those with higher educational attainment; other studies have found higher levels of ACEs in those groups [12,53]. These differences in recall could be various memory biases, such as an experience of maltreatment not being defined as abusive [62]. We also have no external data to validate ACE exposures. Multiple life course influences on individual-level and environmental-levels can change a reproductive plan over time [52,57,63].

For future studies, specific motivations for wanting to have children for future children would yield more insight into the mechanism between ACEs and reproductive plans. Since there was variation by education, future studies on specific social determinants of health to explore the mechanism behind these interactions would be warranted. Following up the GROWH cohort could determine subsequent childbearing and contribute to the literature on reliability between reported intention and fertility behavior. ACE history of mothers would be interesting to assess the intergenerational influence of toxic stress on adolescent pregnancy risk. Future work is needed to fully understand the societal context in which these adverse events occur, which may affect reporting and the proximal mediators in pathways of how ACEs affect health.

Along with being a highly prevalent exposure among women, adverse childhood experiences may have clinical implications on reproductive health trajectories. Because ACEs may influence adolescent pregnancy risk and future pregnancy intentions, taking a trauma-informed care approach on current or previous life adversities may be relevant to help guide discussions about future reproductive health decisions and initiate appropriate clinical and social interventions (e.g., patient counseling, contraceptive education, child protective services) for both pediatric and adult females [31,64]. ACEs come in several different forms, so distinguishing which type of ACE may further determine its severity on reproductive health and possible need for preventative health measures.

## 5. Conclusions

Adverse childhood experiences and specific categories of such early life abuse can influence family planning intentions and potential adolescent pregnancies. Educational attainment demonstrated variation in future reproductive plans and further highlighted the importance of social context in evaluation and intervention.

## Figures and Tables

**Table 1 ijerph-18-00165-t001:** Characteristics of GROWH cohort sample (n = 1482), then stratified by reproductive plans and adolescent pregnancy outcomes. Numbers in parenthesis are % of women with characteristic in each stratum.

Characteristic, n (%)	Sample	Reproductive Plans			Adolescent Pregnancy	
	Total n = 1482	Yes n = 640	Not Sure n = 107	No n = 735	Yes n = 276	No n = 1206
**Age**						
**18–25**	473 (32)	291 (46)	37 (35)	145 (20)	97 (35)	376 (31)
**26–30**	426 (29)	181 (28)	40 (37)	205 (28)	69 (25)	357 (30)
**31–35**	301 (20)	108 (17)	20 (19)	173 (24)	50 (18)	251 (21)
**>35**	279 (19)	58 (9)	10 (9)	211 (29)	58 (22)	221 (18)
**Educational attainment**						
**HS degree or less**	780 (53)	327 (51)	56 (52)	397 (54)	195 (71)	585 (49)
**Above HS degree**	702 (47)	313 (49)	51 (48)	338 (46)	81 (29)	621 (52)
**Household income**						
**<$15,000**	654 (46)	296 (49)	43 (43)	315 (44)	141 (53)	513 (45)
**$15,000–<$35,000**	484 (34)	204 (34)	31 (31)	249 (35)	82 (31)	402 (35)
**≥$35,000**	279 (20)	105 (17)	27 (27)	147 (21)	42 (16)	237 (21)
**Marital Status**						
**Married**	351 (24)	117 (18)	28 (26)	206 (28)	47 (17)	304 (25)
**Living with partner**	224 (15)	120 (19)	20 (19)	84 (12)	45 (16)	179 (15)
**Separated/Widowed/Never married**	898 (61)	401 (63)	58 (55)	439 (60)	183 (67)	715 (60)
**Race**						
**Black**	934 (63)	386 (60)	61 (57)	487 (66)	188 (68)	746 (62)
**White**	409 (28)	186 (29)	37 (35)	186 (25)	71 (26)	338 (28)
**Neither Black nor White**	139 (9)	68 (11)	9 (8)	62 (8)	17 (6)	122 (10)
**Currently Pregnant**						
**Yes**	404 (27)	167 (26)	44 (41)	193 (26)	65 (24)	339 (28)
**Gravidity**						
**Nulligravid**	41 (3)	33 (5)	2 (2)	6 (1)	0 (0)	41 (4)
**Primigravid**	344 (24)	245 (40)	19 (19)	80 (11)	21 (8)	323 (28)
**2**	357 (25)	165 (27)	35 (35)	157 (22)	47 (17)	310 (27)
**3+**	683 (48)	164 (27)	44 (44)	475 (66)	206 (75)	477 (41)
**Parity**						
**Nulliparous**	144 (10)	103 (17)	14 (14)	27 (4)	5 (2)	139 (12)
**Parous**	1279 (90)	495 (83)	85 (86)	699 (96)	270 (98)	1009 (88)
**Trouble getting pregnant**						
**Yes**	157 (12)	103 (19)	10 (10)	44 (7)	25 (10)	132 (12)
**No**	1113 (85)	436 (78)	84 (87)	593 (90)	213 (86)	900 (85)
**Don’t Know**	41 (3)	17 (3)	3 (3)	21 (3)	9 (4)	32 (3)
**Miscarriage history**						
**Yes**	257 (18)	98 (17)	27 (28)	132 (18)	54 (20)	203 (18)
**Depressive symptoms**						
**Yes**	224 (15)	103 (17)	12 (11)	109 (15)	53 (19)	171 (14)
**PTSD-like symptoms**						
**Yes**	120 (9)	55 (9)	4 (4)	61 (9)	32 (13)	88 (8)

Abbreviations: GROWH = Gulf Resilience on Women’s Health, HS = High School, PTSD = Post-Traumatic Stress Disorder.

**Table 2 ijerph-18-00165-t002:** Prevalences of ACEs, adjusted odds ratios between ACEs and reproductive plans, and adjusted odds ratios between ACEs and adolescent pregnancy in GROWH cohort sample (n = 1482).

Exposure	ACEs Prevalences		Wanting Future Children ^1^				Adolescent Pregnancy ^2^	
			Yes (n = 640)		Not Sure (n = 107)		Yes (n = 276)	
**Overall ACE**	n	(%)	OR	(95% CI)	OR	(95% CI)	OR	(95% CI)
**High**	771	(52.0)	1.87	(1.69–2.06) *	1.58	(1.34–1.87) *	1.45	(1.28–1.63) *
**Low**	365	(24.6)	1.31	(1.17–1.47) *	0.93	(0.76–1.14)	1.21	(1.05–1.38) *
**None**	346	(23.4)	1.00	(Ref)	1.00	(Ref)	1.00	(Ref)
**ACE by Age Period**	n	(%)	OR	(95% CI)	OR	(95% CI)	OR	(95% CI)
**Childhood**								
**High**	539	(36.4)	2.04	(1.85–2.25) *	1.95	(1.64–2.31) *	1.30	(1.15–1.46) *
**Low**	522	(35.2)	1.43	(1.30–1.58) *	1.38	(1.16–1.64) *	1.17	(1.05–1.32) *
**None**	421	(28.4)	1.00	(Ref)	1.00	(Ref)	1.00	(Ref)
**Adolescence**								
**High**	517	(34.9)	1.97	(1.79–2.18) *	1.59	(1.34–1.88) *	1.41	(1.25–1.59) *
**Low**	541	(36.5)	1.31	(1.19–1.44) *	1.19	(1.01–1.41) *	1.39	(1.24–1.56) *
**None**	424	(28.6)		(Ref)	1.00	(Ref)	1.00	(Ref)
**ACE Sub-Type**	n	(%)	OR	(95% CI)	OR	(95% CI)	OR	(95% CI)
**Physical Abuse**								
**High**	319	(21.5)	1.91	(1.73–2.10) *	1.72	(1.45–2.03) *	1.21	(1.08–1.36) *
**Low**	354	(23.9)	1.23	(1.12–1.36) *	1.36	(1.16–1.60) *	1.11	(1.00–1.24)
**None**	809	(54.6)	1.00	(Ref)	1.00	(Ref)	1.00	(Ref)
**Emotional Abuse**								
**High**	432	(29.2)	1.99	(1.82–2.17) *	1.16	(0.99–1.36)	0.90	(0.81–1.01)
**Low**	272	(18.4)	1.21	(1.09–1.34) *	1.41	(1.19–1.67) *	0.94	(0.83–1.06)
**None**	778	(52.5)	1.00	(Ref)	1.00	(Ref)	1.00	(Ref)
**Substance Abuse**								
**High**	151	(10.2)	1.74	(1.53–1.97) *	0.80	(0.61–1.04)	1.38	(1.20–1.59) *
**Low**	268	(18.1)	1.24	(1.12–1.38) *	1.47	(1.25–1.73) *	1.08	(0.96–1.22)
**None**	1063	(71.7)	1.00	(Ref)	1.00	(Ref)	1.00	(Ref)
**Sexual Abuse**								
**1+ ACE**	234	(15.8)	1.41	(1.27–1.57) *	0.46	(0.36–0.59) *	1.54	(1.37–1.74) *
**None**	1248	(84.2)	1.00	(Ref)	1.00	(Ref)	1.00	(Ref)
**Neglect**								
**1+ ACE**	323	(21.8)	1.20	(1.10–1.32) *	0.70	(0.59–0.83) *	1.30	(1.17–1.45) *
**None**	1159	(78.2)	1.00	(Ref)	1.00	(Ref)	1.00	(Ref)

^1^ Adjusted for age, race, marital/cohabitation status, highest educational attainment, household income, previous trouble with getting pregnant, history of miscarriage, gravidity, parity, and current pregnancy status. ^2^ Adjusted for age, race, marital/cohabitation status, highest educational attainment, household income, gravidity, parity, current pregnancy status, depressive symptoms, and PTSD-like symptoms. * *p* < 0.05. Abbreviations: ACE = Adverse Childhood Experience, CI = Confidence Interval, GROWH = Gulf Resilience on Women’s Health, OR = Odds Ratio, Ref = Reference.

**Table 3 ijerph-18-00165-t003:** Education-stratified adjusted odds ratios of “Yes” to reproductive plans (n = 1482).

Exposure	Yes to Wanting Future Children ^1^				Interaction *p*-Value
	HS or Below (n = 780)		Above HS (n = 702)		
**Overall ACE**	OR	(95% CI)	OR	(95% CI)	Interaction *p*-value ^2^
**High**	2.40	(2.10–2.75) *	1.40	(1.21–1.61) *	<0.01
**Low**	1.69	(1.45–1.97) *	0.95	(0.80–1.13)	0.04
**None**	1.00	(Ref)	1.00	(Ref)	
**ACE by Age Period**	OR	(95% CI)	OR	(95% CI)	Interaction *p*-value ^2^
**Childhood**					
**High**	2.53	(2.20–2.90) *	1.61	(1.39–1.85) *	0.01
**Low**	1.73	(1.51–1.98) *	1.14	(0.99–1.32)	0.07
**None**	1.00	(Ref)	1.00	(Ref)	
**Adolescence**					
**High**	2.18	(1.90–2.50) *	1.77	(1.52–2.05) *	0.06
**Low**	1.40	(1.23–1.60) *	1.19	(1.03–1.37) *	0.21
**None**	1.00	(Ref)	1.00	(Ref)	
**ACE Sub-Type**	OR	(95% CI)	OR	(95% CI)	Interaction *p*-value ^2^
**Physical Abuse**					
**High**	2.01	(1.75–2.30) *	1.81	(1.56–2.08) *	0.34
**Low**	1.76	(1.54–2.01) *	0.90	(0.79–1.03)	<0.01
**None**	1.00	(Ref)	1.00	(Ref)	
**Emotional Abuse**					
**High**	1.97	(1.74–2.23) *	2.05	(1.79–2.34) *	0.67
**Low**	1.28	(1.11–1.47) *	1.13	(0.97–1.33)	0.24
**None**	1.00	(Ref)	1.00	(Ref)	
**Substance Abuse**					
**High**	1.85	(1.56–2.20) *	1.61	(1.33–1.94) *	0.42
**Low**	1.53	(1.32–1.77) *	1.02	(0.89–1.18)	0.30
**None**	1.00	(Ref)	1.00	(Ref)	
**Sexual Abuse**					
**1+ ACE**	1.57	(1.36–1.81) *	1.25	(1.07–1.45) *	0.22
**None**	1.00	(Ref)	1.00	(Ref)	
**Neglect**					
**1+ ACE**	1.23	(1.09–1.40) *	1.18	(1.03–1.36) *	0.50
**None**	1.00	(Ref)	1.00	(Ref)	

^1^ Adjusted for age, race, marital/cohabitation status, highest educational attainment, household income, previous trouble with getting pregnant, history of miscarriage, gravidity, parity, and current pregnancy status. ^2^
*p*-value for statistical interaction between ACE category and highest educational attainment. * *p* < 0.05. Abbreviations: ACE = Adverse Childhood Experience, CI = Confidence Interval, GROWH = Gulf Resilience on Women’s Health, HS = High School, OR = Odds Ratio, Ref = Reference.

## Data Availability

Data is available to qualified researchers by request of the authors and signing of appropriate data use agreements. Data are not publicly available to protect the privacy of participants.

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
