# Peer review of "Adverse Childhood Experiences on Reproductive Plans and Adolescent Pregnancy in the Gulf Resilience on Women’s Health Cohort"

_ijerph, 2020, doi:10.3390/ijerph18010165_

Round 1

Reviewer 1 Report

Based on the authors' response and resubmitted manuscript, all changes made are accepted.

Author Response

Point 1. Based on the authors' response and resubmitted manuscript, all changes made are accepted.

Response 1. Thank you for reviewing our revision.

Reviewer 2 Report

I am pleased to see that sufficient changes have been made to greatly improve the paper.

Author Response

Response 1. I am pleased to see that sufficient changes have been made to greatly improve the paper.

Response 1. Thank you for reviewing our revision.

Reviewer 3 Report

The paper is much improved.

I am still trying to imagine a process whereby to explain the results.  Perhaps self-esteem theory would be useful.  Everyone wants to feel good about themselves so the question devolves to how to get there.  One route is through education and achievement and work.  Another route is via family and having children.  Perhaps, if one grows up in an abusive environment and has low education, one's options for developing high self-esteem are limited, so that having children is about the only pathway that may seem realistic.  If one was not abused, then one's mental health is probably better and thus also one's job prospects.  If you have higher education, even if abused in the past, one may have a technical skill set that makes one valuable in the marketplace, even with less than optimal mental health. 

Author Response

Point 1. The paper is much improved. I am still trying to imagine a process whereby to explain the results.  Perhaps self-esteem theory would be useful.  Everyone wants to feel good about themselves so the question devolves to how to get there.  One route is through education and achievement and work.  Another route is via family and having children.  Perhaps, if one grows up in an abusive environment and has low education, one's options for developing high self-esteem are limited, so that having children is about the only pathway that may seem realistic.  If one was not abused, then one's mental health is probably better and thus also one's job prospects.  If you have higher education, even if abused in the past, one may have a technical skill set that makes one valuable in the marketplace, even with less than optimal mental health. 

Response 1. Thank you for reviewing our revised manuscript. Based on the above suggestions, we added to the discussion more literature about education and ACEs as well as education and reproductive planning and offered an explanation of our results in the context of job prospects, mental health, and a social-cognitive model of fertility intentions (page 8, lines 251-269).

This manuscript is a resubmission of an earlier submission. The following is a list of the peer review reports and author responses from that submission.

Round 1

Reviewer 1 Report

Thank you for inviting me to review the study entitled “Adverse childhood experiences on reproductive plans & adolescent pregnancy in the Gulf Resilience on Women’s Health Cohort (ijerph-1007630)” related by Flaviano and Harville. The authors evaluated if adverse childhood experiences were associated with reproductive plans or adolescent pregnancy. In addition, sociodemographic variables, pregnancy status, mental health story, age and education levels were analyzed in this study.

In general, the manuscript is written with a logical and coherent structure for the presentation of the study. The aims and results are potentially very interesting, and they were well discussed according to the literature. Comparison with sexual health education would be interesting to understand if reproductive plans and adolescent pregnancy should be affected, addressing sexual debut, neglect, physical/psychological abuse, and sexual abuse.

Reviewer 2 Report

Thank you for the opportunity to review this paper. It is well written, succinct,  the results are well constructed and the discussion is a thoughtful reflection on the results and the possible implications of these for future research, including what improvements might be made in a similar future exercise.

There would be some benefit in providing a definition of the social determinants of health, particularly as these are referred to in the conclusion but are at no point detailed in any way. While education is included in the analysis, employment,income, social support and access to health care are not, therefore the analysis and conclusions requires some qualification on what is meant in relation to the social determinants.

Lines 274-278 identifies some possible concerns associated with lower numbers of ACEs reported by Black participants compared to non-Black participants. The authors might consider that perhaps there is a normalising effect where participants do not view, or internalise, adversity as 'normal' because it is ever present.

There are some minor editorial issues which need to be addressed, as listed below. Otherwise, this is a sound paper and is an interesting contribution to the field in terms of the impact of ACE on reproductive intentions and offers insights into how practitioners may approach discussion with patients with a known history of ACE.

Minor edits:

Line 39: or delayed education (currently reads as educational).

Line 41: there is a missing word/s be reduce with early intervention/support perhaps?

Lines 105-107: this categorisation is not clearly explained?

Line 141: 0 would be better reported as none, which is consistent with the table

Line 220: 'that' is redundant

Line 252: there is a missing word, 'when they accounted for major...'

Line 265: reproductive should be reproduction

LIne 293: there is a missing word, 'help guide discussions about future...'

Line 295: adult females

Line 299: there is a missing word, 'and the need...'

Line 300: missing words, perhaps 'domains of racism and how this affects mental...'

Reviewer 3 Report

This study looks at how childhood adversity impacts early reproduction and fertility intentions. It is an interesting topic and has potential as it is able to differentiate between different ACEs and at different ages (although these are never discussed) however it falls short of publication standards in other areas such as the weak introduction, methodological flaws, in particular not accounting for large amounts of missing data and for not providing a rationale for the research questions or the hypotheses, nor justification for the data used.

Lines 57-58. It is not clear why you make opposite predictions for reproductive intentions and reproductive actualities. One would expect these to go in the same direction, i.e. that ACE is associated with desire for (more) children and earlier pregnancy. In the earlier paragraph (line 43-50) you mention two studies that find less desire for children but these are special cases where women were raising siblings as children already, and women who were HIV-positive which may denote shorter life expectancy which we would expect to be associated with wanting to reproduce, regardless of their childhood experiences. See (Nettle 2010; 2011; Nettle, Coall, and Dickins 2011) for theoretical grounding and empirical support for this hypothesis.

You do go on to find evidence for reproductive intentions in the direction that the theory would expect (positive). Your introduction could do with a stronger rationale that is based on theory and more empirical studies, for instance see (Geronimus 1996; Geronimus, Bound, and Waidmann 1999) for insight into teenage mothers wanting to have earlier births when they live in poor, unhealthy conditions.

Lines 60-69. It is unclear why this particular dataset was used. Is there something special about women affected by the oil spill? Were they chosen because of the oil spill or was this irrelevant to the study? I don’t know how much of a factor the oil spill is, but there might be a link between a serious disaster and changes in fertility behaviour (along the lines of terror-management theory and other evolutionary explanations, or life-history theory) which would have implications for the interpretation of your findings. Please provide a rationale for why these data were chosen.

Line 70 describes sample selection. It is not good practice to include only those who provided valid data on each question because it introduces bias. This is not too bad if there is very little missing data or it can be shown to be missing at random. Table A1 shows that a more than a third of the data are missing. In table A1 (and line 80) you make the claim that there are no differences between missing and non-missing but there is no indicator of significance of the differences.

You should use multiple imputation for missing values on the covariates and control variables (not the outcome and probably not the main ACE indicators). Also if you want to compare between these two samples, they should be identical, in other words only those who answered both questions relating to the outcome variables should be in each model. Although you don’t explicitly link the fertility intentions to the early pregnancy – are they to be treated as two individual sub-studies?

Minor points:

Line 120. Please justify why you use p>0.20 as a cut-off.

Line 36. What do you mean by dose-response? Does this imply causality? If so, please provide references to causal studies or change to ‘association’.

Line 39 is missing a word at the end of the sentence. Also line 41 is missing a word after early.

Line 61 please add “…in 2010” after you name the oil spill.

Bibliography

Geronimus, Arline T. 1996. ‘What Teen Mothers Know’. Human Nature 7 (4): 323–52.

Geronimus, Arline T, John Bound, and Timothy A Waidmann. 1999. ‘Health Inequality and Population Variation in Fertility-Timing’. Social Science & Medicine 49 (12): 1623–36. https://doi.org/10.1016/s0277-9536(99)00246-4.

Nettle, Daniel. 2010. ‘Dying Young and Living Fast: Variation in Life History across English Neighborhoods’. Behavioral Ecology 21 (2): 387–95. https://doi.org/10.1093/beheco/arp202.

———. 2011. ‘Flexibility in Reproductive Timing in Human Females: Integrating Ultimate and Proximate Explanations’. Philosophical Transactions of the Royal Society B: Biological Sciences 366 (1563): 357–65. https://doi.org/10.1098/rstb.2010.0073.

Nettle, Daniel, David Coall, and Thomas E Dickins. 2011. ‘Early-Life Conditions and Age at First Pregnancy in British Women’. Proceedings of the Royal Society B: Biological Sciences 278: 1721–27. https://doi.org/10.1098/rspb.2010.1726.

Reviewer 4 Report

Line 41:  with early ????  It seems like a noun is missing here.

Lines 74, 123:  marital, not martial.

Line 86:  The specific IRB should be cited with approval date, number

Line 120:  why p > .20 is not clear; the intent seems to be to select variables that are significantly associated, so this is confusing to me.

Overall:  The elephant in the living room here may be the size of the family of origin or number of siblings, especially younger siblings.  Larger families may experience greater stress for a host of reasons and that stress may lead to a higher risk of ACEs.  The larger family size may lead to wanting a larger family oneself, which may lead to earlier family formation and lower educational aspirations.  Thus, the linkage found between ACEs and higher fertility intentions might be an artifact of larger family size for the families of origin.  This might be testable if family size was part of the data set. 

Also, it was not clear how many in the sample were nonheterosexuals and if sexual ACEs predicted sexual orientation.  If there were many nonhets then they might need to be analyzed separately or culled out of the sample to make the results clearly apply to heterosexual women.
